# Concave Ni(OH)_2_ Nanocube Synthesis and Its Application in High-Performance Hybrid Capacitors

**DOI:** 10.3390/nano13182538

**Published:** 2023-09-11

**Authors:** Nan Cong, Pan Li, Xuyun Guo, Xiaojuan Chen

**Affiliations:** 1Beijing Academy of Quantum Information Sciences, Beijing 100193, China; congnan@baqis.ac.cn; 2Institute of Analysis and Testing, Beijing Academy of Science and Technology, Beijing 100089, China; cnulp87@126.com; 3Centre for Research on Adaptive Nanostructures and Nanodevices (CRANN) & Advanced Materials Bio-Engineering Research Centre (AMBER), School of Chemistry, Trinity College Dublin, D02PN40 Dublin, Ireland; xguoaa@connect.ust.hk

**Keywords:** Ni(OH)_2_ nanocube, chemical etching, Prussian blue analogs, hybrid capacitor

## Abstract

The controlled synthesis of hollow structure transition metal compounds has long been a very interesting and significant research topic in the energy storage and conversion fields. Herein, an ultrasound-assisted chemical etching strategy is proposed for fabricating concave Ni(OH)_2_ nanocubes. The morphology and composition evolution of the concave Ni(OH)_2_ nanocubes suggest a possible formation mechanism. The as-synthesized Ni(OH)_2_ nanostructures used as supercapacitor electrode materials exhibit high specific capacitance (1624 F g^−1^ at 2 A g^−1^) and excellent cycling stability (77% retention after 4000 cycles) due to their large specific surface area and open pathway. In addition, the corresponding hybrid capacitor (Ni(OH)_2_//graphene) demonstrates high energy density (42.9 Wh kg^−1^ at a power density of 800 W kg^−1^) and long cycle life (78% retention after 4000 cycles at 5 A g^−1^). This work offers a simple and economic approach for obtaining concave Ni(OH)_2_ nanocubes for energy storage and conversion.

## 1. Introduction

Alkaline metal ion batteries and supercapacitors are two primary electric energy stor-age and conversion devices that have been widely applied in electrical vehicles, wearable electronics, and power backups [1,2,3]. To achieve high energy density, high rates, and long cycle life in a single device, hybrid capacitors with a battery-type faradaic cathode and a capacitor-type anode have been rationally designed [4,5,6]. It is well known that the performance of supercapacitors is highly dependent on the chemical composition and morphology of the electrode materials [7]. Transition metal compounds and conducting polymers frequently exhibit a combination of capacitive and faradaic electrochemical properties. Ni(OH)_2_ is the only metallic compound that exhibits faradaic reactions in KOH electrolyte, analogous to battery-type materials [8,9,10]. Ni(OH)_2_ is regarded as a prospective battery electrode material because of its high theoretical capacity, cost effectiveness, and diverse morphologies [11,12,13]. In terms of morphology, nanostructures with a large surface area, low density, and high surface permeability have the potential to significantly improve the efficiency of energy storage devices [14,15]. Therefore, it is essential to exploit materials with suitable porous structures for energy storage devices.

Prussian blue analogs (PBAs) are an intriguing class of crystalline porous materials that have gained considerable attention due to their uniform sizes, abundant types, and flexible compositions [16,17,18,19]. PBA derivatives (such as metal oxides, sulfides, and phosphides) with hollow structures (including frame, box, and porous) have been prepared by chemical corrosion/annealing processes [20,21,22], and showed tremendous potential for energy storage [23,24]. For instance, Chen et al. fabricated crystalline Mn(OH)_2_ and Co(OH)_2_ from PBAs using ion exchange reactions, both of which exhibited outstanding electrochemical properties [25,26]. Although the Ni(OH)_2_ nanocubes derived from PBA have both been reported [27], the chemical etching process is entirely distinct with Ni(OH)_2_ nanocages using Cu_2_O templates [28]. The synthesis of hollow Ni(OH)_2_ from PBA is still unknown [21,29].

In this study, size-controlled concave Ni(OH)_2_ nanocubes were successfully synthesized through a chemical etching method under an ultrasound condition. The synthetic route is illustrated in Figure 1. First, Ni-Co PBA precursors were prepared via a modified self-assembly strategy. The XRD pattern confirmed the precursors as Ni_3_[Co(CN)_6_]_2_·9H_2_O (JCPDS #89-3738). As shown in Appendix A, the precursors exhibited a uniform nanocube shape with a size (side length) of ~400 nm. Then, the PBAs were converted to concave Ni(OH)_2_ nanocubes via an anion exchange reaction in NaOH solution. The corresponding chemical reaction can be described as: Ni_3_[Co(CN)_6_]_2_ (s) + 6OH^−^ (aq) = 3Ni(OH)_2_ (s) + 2[Co(CN)_6_]^3−^ (aq) [15]. The electrochemical measurements indicate that the concave Ni(OH)_2_ nanocubes have a high specific capacity (1624 F g^−1^ at 2 A g^−1^) as well as an exceptional rate capability. Moreover, an optimized hybrid capacitor concave Ni(OH)_2_ nanocube//graphene with high energy density was fabricated.

## 2. Materials and Methods

### 2.1. Synthesis of Ni-Co PBA Nanocubes

A total of 132 mg of C_6_CoK_3_N_6_ was first added into 30 mL of DI water and stirred to make solution A, and 264 mg of C_6_H_5_Na_3_O_7_·2H_2_O and 174 mg of Ni(NO_3_)_2_·6H_2_O were added into 20 mL of DI water and stirred to make a clear green solution, B. Then, solution B was poured into solution A under magnetic stirring for 3 min. The mixed solution was maintained at room temperature for 7 day. After being centrifuged, washed with ethanol numerous times, and vacuum-dried for 24 h, a white-blue powder was achieved.

### 2.2. Synthesis of Concave Ni(OH)_2_ Nanocubes

Ni-Co PBA nanocubes (40 mg) were ultrasonically dispersed into 30 mL of DI water to make a homogenous white-blue suspension. Then, NaOH (150–200 mg) was added. The solution turned light green after 6 h of ultrasonic bathing (40 kHz, 150 W) and mechanical vibration at 40 °C. Finally, the as-prepared products were rinsed with ethanol and vacuum-dried overnight at 80 °C.

### 2.3. Material Characterizations

The structure, chemical composition, and morphology of samples were characterized by XRD (X’Pert Pro MPD system), XPS (ESCALAB MK II), SEM (S4800) equipped with EDS, and TEM (JEM-2200F) equipped with EDS for HRTEM images, SAED patterns, and HAADF-STEM images.

### 2.4. Electrochemical Measurements

The working electrode was prepared by mixing active materials (70%), carbon black (20%), and polyvinylidenefluoride (10%) in the N-methyl-2-pyrrolidinone. Then, the mixture was pressed on nickel foam (NF) (area ~1 cm^2^) and vacuum-dried overnight at 80 °C. The electrochemical property of an individual electrode was measured in a three-electrode system with a 6 M KOH aqueous electrolyte solution, in which a Pt wire and a saturated calomel electrode (SCE) were used as counter and reference electrodes, respectively. The hybrid capacitor electrochemical test was performed in a two-electrode system with concave Ni(OH)_2_ nanocubes pressed on NF as anodes and graphene pressed on NF as cathodes. All electrochemical measurements were carried out by an electrochemical workstation (CHI660D).

## 3. Results

### 3.1. Characterization of Concave Ni(OH)_2_ Nanocubes

An overview SEM image (Figure 2a) reveals that the samples are concave nanocubes with a cavity at each face-center position. The concave nanocubes are approximately 350 nm in size, which is evidently smaller than Ni-Co PBA nanocubes (Appendix A). In addition, the surface of the concave nanocubes is quite rough, and some layered structure on the surface of the concave nanocubes can be observed clearly, as depicted by the red arrows in Figure 2b. This hollow and porous structure increases the specific surface area and shortens the ion transport path, promoting a rapid redox reaction under the conditions of a high scan rate or large current. Figure 2c shows an HRTEM image of the internal lattice structure of a concave Ni(OH_)2_ nanocube, where the interplanar spacing of 0.23 nm corresponds to the (101) plane. The inset of Figure 2c depicts the selected area electron diffraction (SAED) for a concave Ni(OH)_2_ nanocube, in which three diffraction rings are clearly observed and indexed to the (100), (101), and (110) lattice planes of Ni(OH)_2_, indicating its polycrystalline nature. A high-angle annular dark-field scanning transmission electron microscopy (HAADF-STEM) image and the corresponding EDS elemental mapping of Ni and O for nanocages are shown in Figure 2d, indicating a homogeneous distribution of Ni and O elements. Moreover, 100 nm of concave Ni(OH)_2_ nanocubes were also obtained through a similar method, as shown in Appendix A, demonstrating that this ultrasound-assisted etching strategy can be applied to Ni-Co PBA nanocubes with different sizes.

Figure 3a displays the sample’s crystallographic information determined by XRD. The three diffraction peaks appearing at 33.7°, 38.4°, and 51.2° may correspond to the (100), (101), and (102) crystal planes of Ni(OH)_2_ (JCPDS No.14-0117). The broad diffraction peak located at 59°–60.5° may be attributed to the overlapping of the (110) and (003) crystal planes of Ni(OH)_2_ (JCPDS No.14-0117). The peak at 33.7° is slightly higher than the typical Ni(OH)_2_ (JCPDS No.14-0117) at 33.0°, which is likely due to the intercalation of ions during the synthesis process [30]. The atomic ratios of O/Ni for the concave Ni(OH)_2_ nanocubes were determined by EDS to be 2.2 (Figure 3b). The chemical compositions of concave Ni(OH)_2_ nanocubes were also investigated using XPS measurements. Figure 3c shows the high-resolution Ni 2p spectrum of Ni(OH)_2_. The Ni 2p spectrum was fitted with four main peaks and two satellite peaks using a Gaussian–Lorenz fitting method. The fitting peaks at 855.1 eV and 872.8 eV corresponded to Ni^2+^, whereas the peaks at 856.1 eV and 874.1 eV corresponded to Ni^3+^ [31]. Two strong shakeup-type peaks of nickel were detected at 861.3 and 879.4 eV. The analysis of Ni 2p nanomaterials was also added to the revised manuscript [32,33]. Figure 3d depicts three broad XPS spectra of the products at three distinct etching stages: 0 (N1), 1 (N2), and 6 (N3) hours. The peaks at 398.4, 534.6, 782.7, and 858.3 eV in the N1 curve for Ni_3_[Co(CN)_6_]_2_·9H_2_O correspond to the N, O, Co, and Ni element signals, respectively. As the etching reaction progressed from N1 to N3, the peaks at N 1s and Co 2p (denoted by the grey dashed box) diminished and eventually disappeared, whereas the Ni and O peaks (denoted by the red dashed box) are relatively robust. This result indicates that Co and N from [Co(CN)_6_]^3−^ were progressively consumed.

### 3.2. Formation Mechanism of Concave Ni(OH)_2_ Nanocubes

To explore the formation mechanism of concave Ni(OH)_2_ nanocubes, a series of morphological and chemical composition characterizations of intermediates collected from different reaction stages was performed, as shown in Figure 4a–d. When the etching reaction reached 1 h (Figure 4b), all of the sharp corners and edges of the nanocube were dissolved, and a ring island emerged on each face of the nanocube. Arriving at 3 h (Figure 4c), the inner ring island vanished, resulting in the formation of an open hole on each face. Finally, a concave nanocube was discovered (6 h). As the reaction progressed from 0 h to 6 h, the Co/Ni atomic ratio decreased from 2/3 to 0 (Appendix A), indicating that the Co in [Co(CN)_6_]^3−^ was completely replaced by OH^−^. Besides the SEM and EDS analyses, a key chemical composition characterization of intermediates (reaction occurring at 1 h) was carried out. As displayed in Figure 4e–e4, the STEM image and corresponding EDS mapping of the intermediates show that Ni and O elements are uniformly distributed on the nanocube but weak in the face-center region, whereas the Co and N constituents from Ni_3_[Co(CN)_6_]_2_·9H_2_O only appear in the face-center region. Figure 4g depicts the comprehensive distribution of Ni, O, Co, and N along the yellow line (Figure 4f) on the sample. The EDS line results also prove that the edges of the nanocube are converted to Ni(OH)_2_, whereas the face-center regions retain their original composition of Ni_3_[Co(CN)_6_]_2_·9H_2_O.

Based on the above discussions, a possible formation mechanism for concave Ni(OH)_2_ nanocubes is proposed. The edges of Ni-Co PBA nanocubes first undergo ion exchange reactions due to their high curvature and defects. Due to the formation of thick Ni(OH)_2_ layers that protect the interior of the edges from further etching and surfactants that protect the face-center region from dissolution, the etching route shifts to the boundary between the etched edge and the resting face, resulting in the formation of a ring channel on each face. As the channel becomes deeper, the face-center region is completely separated from the structure. After the remaining Ni_3_[Co(CN)_6_]_2_·9H_2_O in the body-center is completely dissolved from six different directions, the concave nanocube structure is formed.

### 3.3. Electrochemical Property of Concave Ni(OH)_2_ Nanocubes

In a standard three-electrode electrochemical setup, the capacity behavior of concave Ni(OH)_2_ nanocubes of approximately 350 nm in size was investigated. Figure 5a shows the CV curves of the Ni(OH)_2_ electrode for various scan rates within the potential window range of 0 to 0.5 V versus SCE. All CV curves exhibit a pair of redox peaks caused by the faradaic reaction Ni(OH)_2_ + OH^−^ ⇄ NiOOH + H_2_O + e^−^ [31,34], implying the battery-type materials of the Ni(OH)_2_ electrode [35,36,37,38]. As the scan rate increases, the anodic and cathodic peaks shift toward higher and lower potentials, respectively, without a significant change in the shape of the CV profiles, revealing that the Ni(OH)_2_ electrode enables rapid redox reactions. Figure 5b presents the typical galvanostatic charge–discharge (GCD) voltage vs. time curves for various current densities. All curves exhibit an asymmetrical shape with well-defined plateaus, implying its battery-type nature. According to the discharge profiles, the specific capacitance of the Ni(OH)_2_ electrode is calculated to be 1624, 1396, 1260, 1196, 1142, 1044, and 980 F g^−1^ at 2, 4, 6, 8, 10, 15, and 20 A g^−1^, respectively (Figure 5c). Even at the current density of 20 A g^−1^, a capacitance of 980 F g^−1^ can be achieved. The Ni(OH)_2_ electrode demonstrates a higher specific capacity than previously reported Ni(OH)_2_ nanostructures, due to the larger Brunauer−Emmett−Teller (BET) surface areas of 187 m^2^ g^−1^, as shown in Appendix A [39,40]. This high-rate capability is attributed to the nanocage structure, which has a greater surface area with a porous structure, allowing ions to move quickly. Figure 5d shows the cycle performance of the Ni(OH)_2_ electrode measured at 10 A g^−1^. After 4000 cycles, the specific capacitance retained almost 77% of the maximum capacitance. Appendix A presents the Nyquist plot of the Ni(OH)_2_ electrode. The equivalent series resistance (ESR) of the Ni(OH)_2_ electrode was evaluated to be 0.52 Ω from the intercept with the real axis.

For practical applications, we constructed a hybrid capacitor device with concave Ni(OH)_2_ nanocubes as the positive electrode and graphene as the negative electrode. As shown in Appendix A, the CV and CP behaviors of graphene were investigated. As expected, it exhibited a characteristic of a double-layer electric capacitor with a specific capacitance of 193 F g^−1^ at 1 A g^−1^. To acquire the maximum capacitance of the hybrid capacitor, the optimal mass ratio is m(concave Ni(OH)_2_ nanocubes)/m(graphene electrodes) = (C^−^ × ΔV^−^)/(C^+^ × ΔV^+^) = 0. 095.

Figure 6a shows the CV curves of the hybrid capacitor measured at a scan rate of 10 mV/s for various voltage windows. The operating potential window can extend to 1.6 V near the voltage of an oxygen evolution reaction in aqueous solution. Figure 6b presents the CV curves of the hybrid capacitor in the potential window of 0–1.6 V at various scan rates. As can be seen, all CV curves have a distorted rectangular shape, indicating that the hybrid device contained both a faradaic capacitor and an electric double-layer capacitance. The voltage–time curves of the hybrid capacitor at varied current densities are depicted in Figure 6c. Figure 6d depicts the calculated specific capacities based on the total mass of the active materials. The maximal specific capacity can reach up to 121 F g^−1^ at 1 A g^−1^. With repeated charge–discharge measurements at a current density of 5 A g^−1^, the cycling stability of the hybrid capacitor was investigated. Figure 6e illustrates the capacity retention variation of the hybrid capacitor along the cycling number, which reveals a capacity loss of 22% after 4000 cycles. The final products are shown in Appendix A, the active materials were partially destroyed and aggregated.

Figure 6f presents the Ragone plot with regard to the energy density and power density of our concave Ni(OH)_2_ nanocubes//graphene hybrid capacitor. The maximal energy density was 42.9 Wh kg^−1^ at a power density of 800 W kg^−1^, and remained at 18.3 Wh kg^−1^ at 8 kW kg^−1^. The energy density of our hybrid capacitor is comparable to that of numerous analogous works, such as flower-like Ni(OH)_2_//AC (25.5 Wh kg^−1^ at 1.28 kW kg^−1^) [41], β-Ni(OH)_2_//AC (36.2 Wh kg^−1^ at 100.6 W kg^−1^) [42], Ni(OH)_2_-ECF//AC/NF (31.3 Wh kg^−1^ at 800 W kg^−1^) [43], NiCo-LDHs/Zn_2_SnO_4_//AC (23.7 Wh kg^−1^ at 284.2 W kg^−1^) [44], and Ni(OH)_2_/graphene//RuO_2_/graphene (48 Wh kg^−1^ at 230 W kg^−1^) [45].

## 4. Conclusions

In summary, we reported concave Ni(OH)_2_ nanocubes through the treatment of Ni-Co PBA nanocubes with NaOH solution in the presence of ultrasound. Different reaction activity between the edges and face-centered regions of Ni-Co PBAs resulted in the formation of a unique hollow structure. When these concave Ni(OH)_2_ nanocubes were used as battery-type electrode materials, they presented high specific capacity and a long cycle life. Moreover, a hybrid capacitor (concave Ni(OH)_2_ nanocubes//graphene) was fabricated and demonstrated outstanding performance with high energy density and good cycling stability. This study provides a simple and efficient method for synthesizing hollow nanomaterials for energy storage applications.

## Figures and Tables

**Figure 1 nanomaterials-13-02538-f001:**
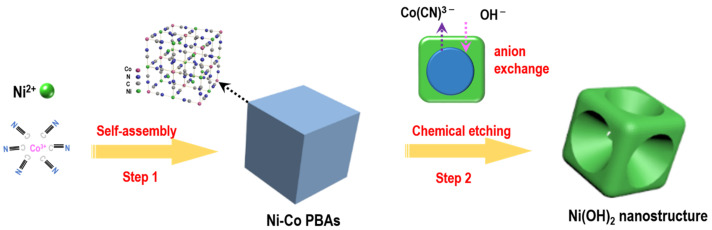
Schematic illustration of the synthesis route of concave Ni(OH)_2_ nanocubes.

**Figure 2 nanomaterials-13-02538-f002:**
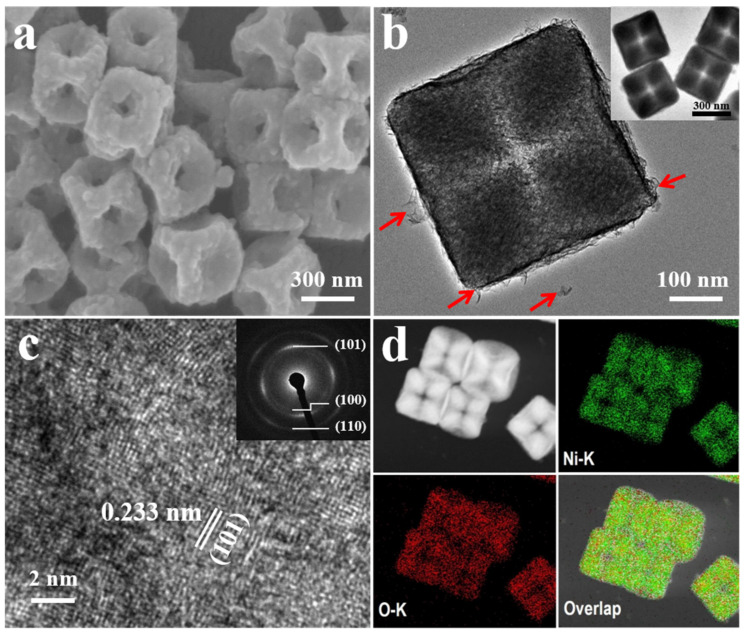
(**a**) SEM image of concave Ni(OH)_2_ nanocubes. (**b**) TEM image of an individual concave Ni(OH)_2_ nanocube, the red arrow marks the layered structure from concave Ni(OH)_2_ nanocubes. (**c**) HRTEM lattice image of a concave Ni(OH)_2_ nanocube with the corresponding SAED pattern image (inset). (**d**) HAADF-STEM image of concave Ni(OH)_2_ nanocubes and the corresponding elemental mapping by STEM.

**Figure 3 nanomaterials-13-02538-f003:**
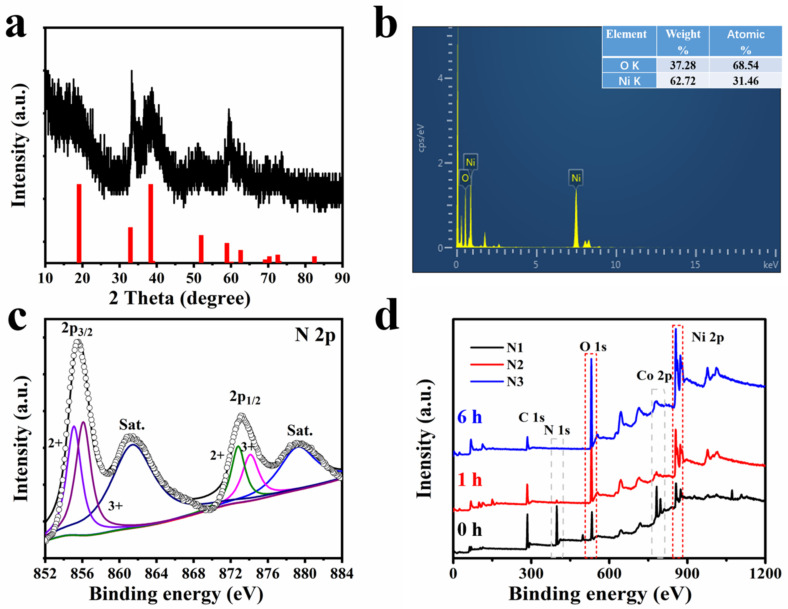
(**a**) The XRD pattern of concave Ni(OH)_2_ nanocubes. (**b**) The EDS spectra of concave Ni(OH)_2_ nanocubes. (**c**) High-resolution XPS spectra of Ni 2p for concave Ni(OH)_2_ nanocubes. (**d**) XPS survey spectra of three samples at different etching stages (0 h, 1 h, and 6 h).

**Figure 4 nanomaterials-13-02538-f004:**
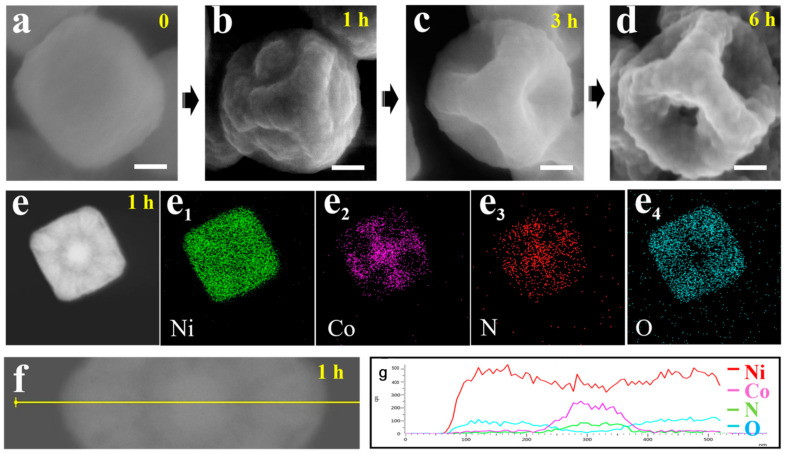
SEM images of the intermediate products at different reaction stages: (**a**) 0 h, (**b**) 1 h, (**c**) 3 h, and (**d**) 5 h. Scale bars: 100 nm. (**e**) STEM image of the samples at 1 h and the corresponding EDX mapping showing (**e1**–**e4**) Ni, Co, N, and O. (**f**) STEM image of the samples at 1 h and the corresponding EDS line scan (**g**) along the yellow line.

**Figure 5 nanomaterials-13-02538-f005:**
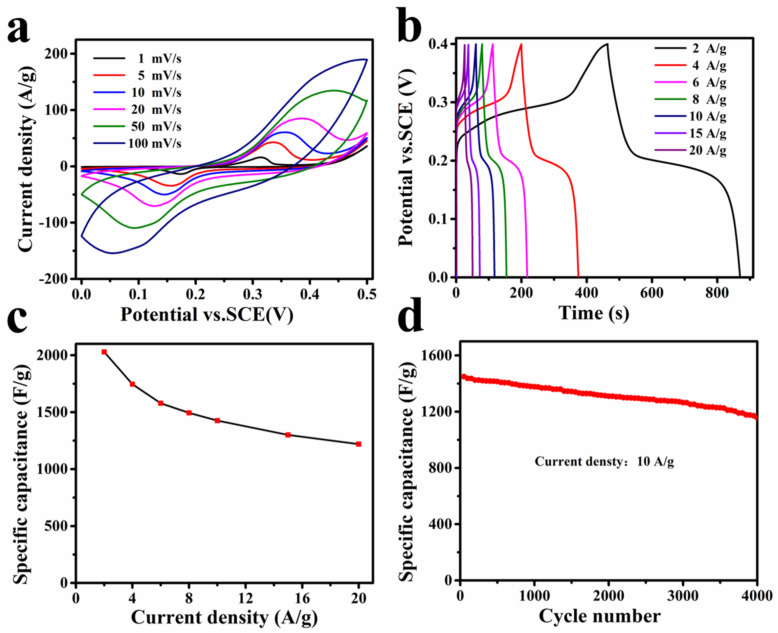
(**a**) CV curves and (**b**) GCD curves of concave Ni(OH)_2_ nanocubes. (**c**) Specific capacity of concave Ni(OH)_2_ nanocubes at different current densities. (**d**) Cycle performance of concave Ni(OH)_2_ nanocubes at 10 A/g.

**Figure 6 nanomaterials-13-02538-f006:**
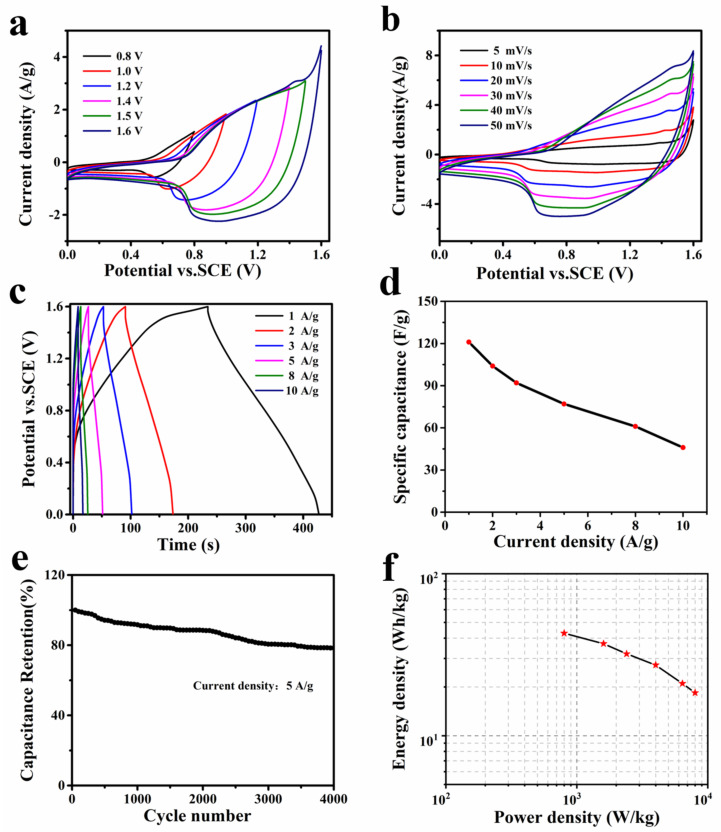
(**a**) CV curves of the concave Ni(OH)_2_ nanocubes//graphene hybrid capacitor with varying potential windows (at 10 mV s^−1^). (**b**) CV and (**c**) GCD curves of the hybrid capacitor. (**d**) Specific capacity of the hybrid device as a function of current densities. (**e**) Cycle performance measured at 5 A g^−1^ for 4000 cycles. (**f**) Ragone plots of the hybrid capacitor.

## Data Availability

Not applicable.

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
