# Peer review of "Concave Ni(OH)2 Nanocube Synthesis and Its Application in High-Performance Hybrid Capacitors"

_nanomaterials, 2023, doi:10.3390/nano13182538_

Round 1
Reviewer 1 Report
I have reviewed the submitted manuscript entitled “Concave Ni(OH)2 nanocube synthesis and its application in high-performance hybrid capacitors” by X. Chen et al submitted to Nanomaterials/MDPI.
In recent decades, energy storage systems such as rechargeable batteries and hybrid supercapacitors have received great attention, especially in industries like hybrid electric vehicles and smart portable electronics. Due to their excellent performance, the reported Ni(OH)2-based materials have been extensively studied for application in pseudocapacitors. However, the authors in this work submitted to Nanomaterials (MDPI) have considered fabricating concave Ni(OH)2 nanocubes using an etching strategy. The purpose of this work is to create void space and transport charges more efficiently while also contributing to electro-capacitance using porous structures. The level of originality is a bit questionable unless otherwise the authors substantially address this, the current version does not warrant publication. This reviewer means to say, it is worthless.
Apart from this, the work itself is well presented, and the manuscript is OK with reasonable electrochemical data drawn from the capacitor results. The work in its present form is publishable but needs some revisions before rendering a final decision.
The following points need to be considered.
1. The concept of porous Ni(OH)2 nanocube and its high electrochemical performance for supercapacitors has been well known and reported in Langmuir (doi.org/10.1021/acs.langmuir.7b03035) using Ni–Co Prussian blue analog (PBA) as a precursor. Being this is the case, what is the originality of the current work? The work resembles very similar.
2. The hierarchical core-shell Ni(OH)2 reported in doi.org/10.1016/j.jcis.2022.08.057 has exceeded the capacitance value reported for concave, justify.
3. The section of the introduction must emphasize the original idea of this work.
4. The shape of the CV curve and its redox reactions must be stated in Section 3.3.
5. The electrochemistry (CV, Nyquist plots) of the reported work and their capacitance values (Symmetric and Asymmetric) must be benchmarked to the one reported by the relevant materials published by Manickam Minakshi et al and the group.
6. The distorted rectangular shape of Ni(OH)2 must be explained appropriately.
7. How the values (c g-1) for the hybrid capacitor are calculated? What is the mass of the material?
The language is OK.
Author Response
Dear reviewers and editor,
The response to the referees is attached in the file"author-coverletter-31562838.v1.pdf"

Reviewer 2 Report
The author describes the “Concave Ni(OH)2 nanocube synthesis and its application in high-performance hybrid capacitors”. This original article is quite interesting from a technological point of view. The author should revise their manuscript based on the comments and suggestions. I recommended a Major revision of the manuscript.
The Major suggestion below:
- The XRD pattern does not match the Ni(OH)2 phase of JCPDS No.14-0117. The author should recheck the XRD pattern of the prepared materials.
- In the abstract the author can state a clear research question to convey the main objective of this study.
- The author should provide the Raman spectra of all the prepared samples.
- The author should index the peaks in the XRD pattern in Fig.S1 b.
- The figure quality is too poor and the author should improve the quality of the figure.
- The author should re-check the XPS spectra of the samples.
- The author should provide the 10000 cycle stability of the Ni(OH)2 nanocubes for a better understanding of the long durability of the material.
- For the complete understanding of the supercapacitor mechanism by Ni(OH)2 nanocubes and the Ni(OH)2 nanocubes/graphene hybrid capacitor, post morphology and compositional changes can be studied with XPS and FESEM analysis.
- The author should provide the 10000 cycle stability of the Ni(OH)2 nanocubes/graphene for a better understanding of the long durability of the material.
- The author should provide the after-stability EIS measurement of all samples in the revised manuscript.
- The author should improve the grammatical and typo errors in the paper.
Minor editing of the English language is required.
Author Response
Dear reviewers and editor,
The response to the referees is attached in the file"author-coverletter-31598798.v1.pdf"

Reviewer 3 Report
The manuscript entitled "Concave Ni(OH)2 Nanocube Synthesis and Its Application in High-Performance Hybrid Capacitors" presents a fascinating and highly valuable research topic in the field of supercapacitors. The authors have prepared concave Ni(OH)2 nanocubes using an ultrasound-assisted chemical etching strategy. The results presented in this manuscript are very interesting and thorough. The analytical characterizations and morphological studies are also very thorough and comprehensive. The performance of the Ni(OH)2 nanocubes is very impressive.
After careful revision, I recommend the manuscript for publication after addressing the following minor points:
- The authors should provide stability studies of the Ni(OH)2 using PXRD. This is important because the decomposed phase can also exhibit excellent capacitance performance.
- The authors should present the surface analysis of the Ni(OH)2, including BET and N2 sorption isotherms. This information can be found in the following articles and cite them accordingly. (Advanced Energy Materials 6 (13), 1600110, Small 14 (37), 1801233 and Advanced Energy Materials 6 (24), 1601189)
- It would be interesting to see a literature comparison of all the Ni(OH)2-based capacitors. This would help to contextualize the performance of the nanocubes presented in this manuscript.
- The authors should also provide a cost comparison of their approach. This would help to demonstrate the economic viability of their method.
Author Response
Dear reviewers and editor,
The response to the referees is attached in the file"author-coverletter-31598754.v1.pdf"

Round 2
Reviewer 1 Report
The revised version of the manuscript has been significantly improved in quality and the ambiguities are resolved. Therefore, in this reviewer's opinion, this version is suitable for publication.
Reviewer 2 Report
The authors have carried out all my comments. So I recommend acceptance of this paper.